# On the Complexity of Identification in Linear Structural Causal Models

**Julian Dörfler**[*]
Saarland University

**Benito van der Zander**[*]
University of Lübeck

**Markus Bläser**[†]
Saarland University

**Maciej Liśkiewicz**[†]
University of Lübeck

## Abstract

Learning the unknown causal parameters of a linear structural causal model is a fundamental task in causal analysis. The task, known as the problem of identification, asks to estimate the parameters of the model from a combination of assumptions on the graphical structure of the model and observational data, represented as a non-causal covariance matrix. In this paper, we give a new sound and complete algorithm for generic identification which runs in polynomial space. By a standard simulation result, namely $\mathsf{PSPACE} \subseteq \mathsf{EXP}$, this algorithm has exponential running time which vastly improves the state-of-the-art double exponential time method using a Gröbner basis approach. The paper also presents evidence that parameter identification is computationally hard in general. In particular, we prove, that the task asking whether, for a given feasible correlation matrix, there are exactly one or two or more parameter sets explaining the observed matrix, is hard for $\forall\mathbb{R}$, the co-class of the existential theory of the reals. In particular, this problem is coNP-hard. To our best knowledge, this is the first hardness result for some notion of identifiability.

## 1 Introduction

Recognizing and predicting the causal effects and distinguishing them from purely statistical correlations is an important task of empirical sciences. For example, identifying the causes of disease and health outcomes is of great significance in developing new disease prevention and treatment strategies. A common approach for establishing causal effects is through randomized controlled trials (Fisher, [20]) – called the gold standard of experimentation – which, however, requires physical intervention in the examined system. Therefore, in many practical applications, experimentation is not always possible due to cost, ethical constraints, or technical feasibility. E.g., to learn the effects of radiation on human health one cannot conduct interventional studies involving human participants. In such cases, a researcher may use an alternative approach and establish cause-effect relationships by combining existing observed data with the knowledge of the structure of the system under study. This is called the problem of *identification* in causal inference (Pearl, [32]) and the approach is commonly used in various fields, including modern ML.

A key ingredient of this framework is the way the underlying structure models the true mechanism behind the system. In general, this is done using structural causal models (SCMs) [32, 3]. In this work, we focus on the problem of identification in linear SCMs, known also as structural equation models (SEMs) [7, 18]. They represent the causal relationships between observed random variables

---

[*]Equal contribution as first authors.
[†]Equal contribution as last authors.

38th Conference on Neural Information Processing Systems (NeurIPS 2024).

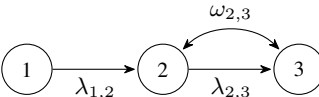

Figure 1: An IV example.

assuming that each variable $X_i$, with $i = 1, \ldots, n$ is linearly dependent on the remaining variables and an unobserved error term $\varepsilon_i$ of normal distribution with zero mean: $X_j = \sum_i \lambda_{i,j} X_i + \varepsilon_j$. The model implies the existence of some covariance matrix $\Omega = (\omega_{i,j})$ between the error terms. In this paper, we consider mainly *recursive models*, i.e., we assume that, for all $i > j$, we have $\lambda_{i,j} = 0$, nonetheless we discuss how our methods can be extended to the general case.

Linear SCMs can be represented as a graph with nodes $\{1, \ldots, n\}$ corresponding to variables $X_1, \ldots, X_n$ and with directed and bidirected edges. A directed edge $i \to j$ represents a linear influence $\lambda_{i,j} \neq 0$ of a parent node $i$ on its child $j$. A bidirected edge $i \leftrightarrow j$ represents a correlation $\omega_{i,j} \neq 0$ between error terms $\varepsilon_i$ and $\varepsilon_j$ (cf. Figure 1).

Writing the coefficients of all directed edges as an adjacency matrix $\Lambda = (\lambda_{i,j})$ and the coefficients of all bidirected edges as an adjacency matrix $\Omega = (\omega_{i,j})$, the covariances $\sigma_{i,j}$ between each pair of observed variables $X_i$ and $X_j$ can be calculated as matrix $\Sigma = (\sigma_{i,j})$:

$$\Sigma = (I - \Lambda)^{-T} \Omega (I - \Lambda)^{-1}, \tag{1}$$

where $I$ is the identity matrix [21]. Knowledge of the parameters $\lambda_{i,j}$ allows for the prediction of all causal effects in the system. The key task here is to learn $\Lambda$ from the observed covariances $\Sigma$ assuming $\Omega$ remains unknown. This leads to the formulation of the identification problem in linear SCMs as solving for the parameter $\Lambda$ using equation (1). If the problem asks to find symbolic solutions involving only symbols $\sigma_{i,j}$, we call it the problem of *symbolic identification*. In the case when the parameter can be determined uniquely almost everywhere using $\Sigma$ alone, we call the instance to be *generically identifiable* (for a formal definition, see Sec. 2). If the goal is to find, for a given $\Sigma$ of rational numbers, the numerical solutions, we call it the problem of *numerical identification*. In this paper, we study the computational complexity of both variants of the problem.

**Previous Work.** The identification in linear SCMs and its applications have been a subject of research interest for many decades, including the early work in econometrics and agricultural sciences [42, 41, 19, 8]. Currently, it seems, that one of the most challenging tasks in this field is providing efficient computational methods to find solutions, both for symbolic and for numeric variants, or providing evidence that the problems are computationally intractable.

The generic identification can be computed using standard algebraic tools for solving symbolic polynomial equations (1). Such an approach provides a *sound* and *complete* method, i.e., it is guaranteed to identify all identifiable instances. However, common algorithms for solving such equations usually use Gröbner basis computation, whose time complexity is doubly exponential in the worst case [22]. So far, it has remained widely open whether the double exponential function is a sharp upper bound on the computational complexity of the generic identifiability.

Most approaches to solving the problem in practice are based on instrumental variables, in which the causal direct effect is identified as a fraction of two covariances [41, 8]. For example, in the linear model shown in Figure 1, one can calculate first $\lambda_{1,2} = \sigma_{1,2}$ and then $\lambda_{2,3} = \frac{\lambda_{1,2}\lambda_{2,3}}{\lambda_{1,2}} = \frac{\sigma_{1,3}}{\sigma_{1,2}}$. The variable $X_1$ is then called an instrumental variable (IV). This method is sound but not complete, that is, when it identifies a parameter, then it is always correct. However, when the method fails due to a missing IV, then the parameter might still be identified by other means. This approach has inspired intensive research aimed at providing computational methods that may not be complete but enable efficient algorithms and identify a significantly large number of cases.

Conditional IVs (cIVs) are one of the most natural extensions of simple IVs [8, 31]. The corresponding identification method is based on an efficient, polynomial time algorithm for finding conditional IVs [38]. More complex criteria and methods proposed in the literature, which are also accompanied by polynomial time algorithms, involve instrumental sets (IS) [10] half-treks (HTC) [21], instrumental cutsets (ICs) [28], auxiliary instrumental variables (aIVs) [15]. The generalized HTC (gHTC) [13, 40] and auxiliary variables (AVS) [13, 14] can be implemented in polynomial time provided

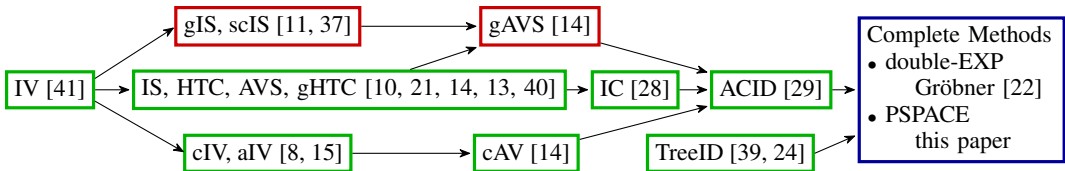

Figure 2: Methods for generic identification in linear SCMs. An arrow from methods $A \to B$ means that $B$ subsume all methods $A$, i.e., any instance that can be identified by any of methods $A$ can be identified by method $B$ and this inclusion is proper. Green boxes mean there exist polynomial-time algorithms to apply the method, a red box means no such algorithm is known or the method has been proven to be NP-hard. The blue box includes the complete methods.

that the number of incoming edges to each node in the causal graph is bounded. The methods based on generalized IS (gIS), a simplified version of the criterion (scIS), and generalized AVS (gAVS) appeared to be computationally intractable [10, 9, 11, 37, 14]. The auxiliary cutsets (ACID) algorithm [29] subsumes all the above methods in the sense that it covers all the instances identified by them. Recently [39, 24] provide the TreeID algorithm for identification in tree-shaped linear models. TreeID is incomparable since it is complete for the subclass of tree-like SCMs. However, TreeID does not work for general SCMs, which is the focus of our work. Figure 2 summarizes these results.

Numerical parameter identification, known in the literature as the estimation of the parameters of structural equation models, has been the subject of a considerable amount of research, which has resulted in significant progress in theoretical understanding and development of estimation methods [2, 26, 12, 30, 7, 25]. Currently, in practical applications (e.g. in econometrics, psychometrics, or biometrics), methods based on maximum likelihood (ML) or generalized least squares estimator (GLS) are commonly used to find model parameters. However, despite the great importance of this problem and considerable effort in method development, the computational complexity of the parameter estimation problem remains unexplored. In our work, we provide, to our knowledge, the first hardness result for a very basic variant of the SEM parameter estimation problem, which we call numerical identification.

**Our Contribution.** We improve significantly the best-known upper bound on the computational complexity for generic identification and provide evidence that parameter identification is computationally hard in general. In more detail, our contributions are as follows:

- We provide a polynomial-space algorithm for sound and complete generic parameter identification in linear models. This gives an exponential running time which vastly improves the state-of-the-art double exponential time method using Gröbner basis. In our approach, we formulate generic identifiability as a formula over real variables with both existential and universal quantifiers and then use Renegar's algorithm [33].

- Our constructive technique allows us to prove an $\exists \forall \mathbb{R}$ (and $\forall \exists \mathbb{R}$) upper bound on generic identifiability, for the well-studied complexity class $\exists \forall \mathbb{R}$ (see Sec. 2.2 for definitions). It is an intermediate class between NP and PSPACE.

- We prove that numerical identification is hard for the complexity class $\forall \mathbb{R}$. In particular, the problem is coNP-hard. Our complexity characterization is quite precise since we show a (promise) $\forall \mathbb{R}$ upper bound for numerical identification. To the best of our knowledge, this is the first hardness result for some notion of identifiability.

- On the other hand, we show that numerical identifiability can be decided in polynomial space.

- If an instance is non-identifiable, then an important task is to identify as many model parameters as possible. We are particularly interested in identifying the parameters of specific edges in the graph representing the linear model. In the paper, we obtain, for "edge identifiability", the same results as for the common identifiability problem. Since these proofs are essentially the same, they can be found in Appendix A.

## 2 Preliminaries

### 2.1 The Problems of Identification in Linear Causal Models

A mixed graph is a triple $G = (V, D, B)$ where $V := \{1, \ldots, n\}$ is a finite set of nodes and $D \subseteq V \times V$ and $B \subseteq \binom{V}{2}$ are two sets of directed and bidirected edges, respectively. Let $\mathbb{R}^D$ be the set of matrices $\Lambda = (\lambda_{i,j}) \in \mathbb{R}^{n \times n}$ with $\lambda_{i,j} = 0$ if $i \to j$ is not in $D$ and let $\mathrm{PD}(n)$ denote the cone of positive definite $n \times n$ matrices. Let $\mathrm{PD}(B)$ be the set of matrices $\Omega = (\omega_{i,j}) \in \mathrm{PD}(n)$ with $\omega_{i,j} = 0$ if $i \neq j$ and $i \leftrightarrow j$ is not an edge in $B$. For now, we will only consider recursive models, i.e. we assume that, for all $i > j$, we have $\lambda_{i,j} = 0$ (in Sec. 7 we will discuss how our methods can be extended to general graphs allowing cycles). Thus, the directed graph $(V, D)$ accompanied with the model is acyclic. We will assume w.l.o.g. that the nodes are topologically sorted, i.e., there are no edges $i \to j$ with $i > j$.

Denote by $\mathcal{N}_n(\mu, \Sigma)$ the multivariate normal distribution with mean $\mu \in \mathbb{R}^n$ and covariance matrix $\Sigma$. The linear SCMs $\mathcal{M}(G)$ associated with $G = (V, D, B)$ is the family of multivariate normal distributions $\mathcal{N}_n(0, \Sigma)$ with $\Sigma$ satisfying equation (1), for $\Lambda \in \mathbb{R}^D$ and $\Omega \in \mathrm{PD}(B)$. A model in $\mathcal{M}(G)$ is specified in a natural way in terms of a system of linear structural equations: $X_j = \sum_{i \in \mathrm{pa}(j)} \lambda_{i,j} X_i + \varepsilon_j$, for $j = 1, \ldots, n$, where $\mathrm{pa}(j)$ denote the parents of $j$ in $G$. If $\varepsilon = (\varepsilon_1, \ldots, \varepsilon_n)$ is a random vector with the multivariate normal distribution $\mathcal{N}_n(0, \Sigma)$ and $\Lambda \in \mathbb{R}^D$, then the random vector $X = (X_1, \ldots, X_n)$ is well defined as a solution to the equation system and follows a centered multivariate normal distribution with covariance matrix $(I - \Lambda)^{-T} \Omega (I - \Lambda)^{-1}$ (see, e.g. [17]).

For a given (acyclic) mixed graph $G = (V, D, B)$, define the parametrization map

$$\phi_G : (\Lambda, \Omega) \mapsto (I - \Lambda)^{-T} \Omega (I - \Lambda)^{-1}$$

and let $\Theta := \mathbb{R}^D \times \mathrm{PD}(B)$. We say that $G$ is *globally* identifiable if $\phi_G$ is injective on $\Theta$ [17].

Global identification can be decided easily, see [17, Theorem 2]. However, it is a very strong property. For instance, as seen in the introduction, in Figure 1, we can recover the parameter $\lambda_{2,3}$ as $\frac{\sigma_{1,3}}{\sigma_{1,2}}$. If $\sigma_{1,2} = 0$, then the identification fails, so the instance is not globally identifiable. But identification fails only in the (very unlikely) case that $\sigma_{1,2} = 0$. This leads to the concept of *generic identifiability*:

**Definition 1** (Generic Identifiability, [21]). *The mixed graph $G$ is said to be* generically *identifiable if $\phi_G$ is injective on the complement $\Theta \setminus \mathcal{V}$ of a proper (i.e., strict) algebraic subset $\mathcal{V} \subset \Theta$.*

Given matrices $\Lambda_0 \in \mathbb{R}^D$ and $\Omega_0 \in \mathrm{PD}(B)$, the corresponding *fiber* is defined by

$$\mathcal{F}_G(\Lambda_0, \Omega_0) = \{(\Lambda, \Omega) \mid \phi_G(\Lambda, \Omega) = \phi_G(\Lambda_0, \Omega_0), \Lambda \in \mathbb{R}^D, \Omega \in \mathrm{PD}(B)\}.$$

A fiber contains all pairs of matrices that induce the same observed covariance matrix $\Sigma$. For $\Sigma \in \mathrm{im}\,\phi_G$, we also write $\mathcal{F}_G(\Sigma)$ for the fiber belonging to $\Sigma$. We can phrase identifiability in terms of fibers:

- $G$ is globally identifiable, if $|\mathcal{F}_G(\Lambda_0, \Omega_0)| = 1$ for all $\Lambda_0 \in \mathbb{R}^D$ and $\Omega_0 \in \mathrm{PD}(B)$.
- $G$ is generically identifiable, if $|\mathcal{F}_G(\Lambda_0, \Omega_0)| = 1$ for Zariski almost all $\Lambda_0 \in \mathbb{R}^D$ and $\Omega_0 \in \mathrm{PD}(B)$.

Generic identifiability asks whether all parameters are almost always identifiable in the Zariski sense, that is, everywhere except for a lower dimensional algebraic set. For generic identifiability, we only consider the parameters $\lambda_{i,j}$ since the parameters $\omega_{k,l}$ can be recovered from the $\lambda_{i,j}$ and $\sigma_{i,j}$ using (1), see also [16].

It is also of interest to ask whether a single parameter $\lambda_{i,j}$ is almost always identifiable. For this, we consider the projection of the fiber on the single parameter, which we will also call an *edge fiber*:

$$\mathcal{F}_G^{i,j}(\Lambda_0, \Omega_0) = \{\Lambda_{i,j} \mid (\Lambda, \Omega) \in \mathcal{F}_G(\Lambda_0, \Omega_0)\}.$$

(Above $\Lambda_{i,j}$ denotes the entry of $\Lambda$ in the position $(i, j)$, that is, $\lambda_{i,j}$.)

**Definition 2.** *The parameter $\lambda_{i,j}$ is generically edge identifiable, if $|\mathcal{F}_G^{i,j}(\Lambda_0, \Omega_0)| = 1$ for Zariski almost all $\Lambda_0 \in \mathbb{R}^D$ and $\Omega_0 \in \mathrm{PD}(B)$.*

Global and generic identifiability are properties of the given mixed graph. In this work, we also study identification as a property of the observed numerical data, i.e., of the observed covariance matrix $\Sigma$.

**Definition 3** (Numerical Identifiability). *Given an acyclic mixed graph $G = (V, D, B)$ and a feasible matrix $\Sigma$, decide whether the parameters are uniquely identifiable, i.e. if $|\mathcal{F}_G(\Sigma)| = 1$?*

Note that this is a promise problem. We assume that $\Sigma$ is feasible, i.e., in the image of $\phi_G$. Therefore, we shall also study the feasibility problem: Given $\Sigma$, is it contained in $\operatorname{im} \phi_G$?

Similarly we can also define numerical edge identifiability: For a given feasible $\Sigma$, test whether the edge fiber $\Sigma$ belongs to has size 1 or $> 1$.

## 2.2 The (Existential) Theory of the Reals

The existential theory of the reals (ETR) is the set of true sentences of the form

$$\exists x_1 \ldots \exists x_n \; \varphi(x_1, \ldots, x_n), \tag{2}$$

where $\varphi$ is a quantifier-free Boolean formula over the basis $\{\vee, \wedge, \neg\}$ and a signature consisting of the constants $0$ and $1$, the functional symbols $+$ and $\cdot$, and the relational symbols $<$, $\leq$, and $=$. The sentence is interpreted over the real numbers in the standard way. The theory forms its own complexity class $\exists \mathbb{R}$ which is defined as the closure of ETR under polynomial-time many-one reductions. Many natural problems have been shown to be complete for ETR, for instance the computation of Nash equilibria [35], the famous art gallery problem [1], or training neural networks [5], just to mention a few. See the recent compendium [34] for a complete overview.

It turns out that one can simplify the form of an ETR-instance. We can get rid of the relations $<$ and $\leq$ and it is sufficient to consider only Boolean conjunctions. More precisely, the following problem is $\exists \mathbb{R}$-complete: Given polynomials $p_1, \ldots, p_m$ in variables $x_1, \ldots, x_n$, decide whether there is a $\xi \in \mathbb{R}^n$ such that

$$p_1(\xi) = \cdots = p_m(\xi) = 0. \tag{3}$$

By Tseitin's trick, we can assume that all polynomials are of one of the forms

$$ab - c, \; a + b - c, \; a - b, \; a - 1, \; a \tag{4}$$

and all variables in each of the polynomials are distinct. Note that all polynomials in (4) have degree at most two. Therefore, this problem is also called the feasibility problem of quadratic equations QUAD. For a proof, see e.g. [35].

**Universal Quantification.** If, instead of considering existentially quantified true sentences, we consider universally quantified true sentences of the form

$$\forall x_1 \ldots \forall x_n \; \varphi(x_1, \ldots, x_n), \tag{5}$$

where $\varphi$ is again a quantifier-free Boolean formula, and form the closure under polynomial-time many-one reductions, we obtain the complexity class $\forall \mathbb{R}$. Using De Morgan's law, it is easy to see the well-known fact that $\forall \mathbb{R} = \mathsf{co}\text{-}\exists \mathbb{R}$, i.e. it is the complement class of $\exists \mathbb{R}$.

It is also possible to alternate quantifiers, giving rise to a whole hierarchy, comparable to the well-known polynomial time hierarchy, see [36]. We call the corresponding classes $\exists \forall \mathbb{R}$, $\forall \exists \mathbb{R}$, $\ldots$

**Complexity of $\exists \mathbb{R}$ and $\forall \mathbb{R}$.** It is easy to see that quantification over real variables can be used to simulate quantification over Boolean variables by adding the constraint $x(x - 1) = 0$. This way we can convert 3SAT-formulas to ETR-formulas, proving the well-known containment $\mathsf{NP} \subseteq \exists \mathbb{R}$.

With his celebrated result about quantifier elimination, Renegar [33] proved that the truth of any sentence over the reals with a constant amount of quantifier alternations is decidable in PSPACE. This in particular implies

$$\mathsf{NP} \subseteq \exists \mathbb{R} \subseteq \mathsf{PSPACE} \qquad \text{and} \qquad \mathsf{coNP} \subseteq \forall \mathbb{R} \subseteq \mathsf{PSPACE}. \tag{6}$$

While all these inclusions are believed to be strict, it is unknown for all of them.

## 3 Finding Another Solution

Numerical identification is a promise problem, i.e., we assume that the given input is feasible. Being a promise problem means that an algorithm for numerical identification should output the correct answer whenever the input is feasible. But it can output anything when the input is not feasible. We give some further information about promise problems in Appendix B for the reader's convenience.

For our hardness proof, we need to look at instances of ETR or QUAD that are satisfiable. Of course, deciding whether a satisfiable instance is satisfiable is a trivial task. So the task will be to decide whether the satisfiable instance has another solution. We call the corresponding promise problems $\text{ETR}^{++}$ and $\text{QUAD}^{++}$.

It turns out that these promise problems are $\exists \mathbb{R}$-hard. Since $\text{QUAD}^{++}$ is a special case of $\text{ETR}^{++}$, it suffices to prove this for $\text{QUAD}^{++}$. Let $y$ be an extra variable. We will plant an extra solution into the system (3):

$$y(y-1) = 0 \tag{7}$$
$$yx_i = 0 \qquad i = 1, \ldots, n \tag{8}$$
$$(y-1)p_j = 0 \qquad j = 1, \ldots, m \tag{9}$$

**Lemma 1.** *The system above has the following solutions:*

1. *$y = 1$, $x_1 = \cdots = x_n = 0$*

2. *$y = 0$, $x_1 = \xi_1, \ldots, x_n = \xi_n$, where $\xi \in \mathbb{R}^n$ is any solution to the original instance.*

*In particular, the system always has a solution. It has more than one solution iff the original* QUAD*-instance is satisfiable.*

*Proof.* The first equation (7) constrains $y$ to be $\{0, 1\}$-valued. If $y = 0$, then the equations (8) are trivially satisfied and (9) reduces to the original instance (3). If $y = 1$, then the equations (9) are trivially satisfied and (8) reduces to $x_1 = \cdots = x_n = 0$. Note that in both cases we always get different solutions since the $y$-value differs. $\square$

Using the transformation in the lemma above, we can map any QUAD-instance into a $\text{QUAD}^{++}$-instance and obtain

**Corollary 1.** $\text{ETR}^{++}$ *and* $\text{QUAD}^{++}$ *are* $\exists \mathbb{R}$*-hard.*

## 4 Hardness of Numerical Identifiability

This section is dedicated to proving:

**Theorem 2.** *Numerical identifiability is* $\forall \mathbb{R}$*-hard.*

The proof consists of building a polynomial-time reduction from the complement of $\text{QUAD}^{++}$ to numerical identifiability, i.e., we construct an acyclic mixed graph $G$ and a $\Sigma \in \text{im}\,\phi_G$, such that the fiber $\mathcal{F}_G(\Sigma)$ has size 1 iff the given $\text{QUAD}^{++}$-instance has only one solution. For this, we use the following characterization of fibers due to [16]:

**Lemma 3.** *Let $G = (V, D, B)$ be an acyclic mixed graph, and let $\Sigma \in \text{im}\,\phi_G$. The fiber $\mathcal{F}_G(\Sigma)$ is isomorphic to the set of matrices $\Lambda \in \mathbb{R}^D$ that solve the equation system*

$$[(I - \Lambda)^T \Sigma (I - \Lambda)]_{i,j} = 0, \qquad i \neq j, i \leftrightarrow j \notin B \tag{10}$$

We construct $G$ as follows: The directed edges form a bipartite graph with edges going from the bottom layer to the top layer. Every node at the bottom layer has outdegree one. Moreover bidirected edges exist between all pairs of nodes, except for certain pairs of nodes of the top layer. See Figure 3 (left-hand side) for an illustration.

This missing edge in Figure 3 corresponds to the equation

$$0 = \sigma_{i,j} - \sum_{\ell=1}^{n} \sigma_{a_\ell,j} \lambda_{a_\ell,i} - \sum_{k=1}^{m} \sigma_{b_k,i} \lambda_{b_k,j} + \sum_{\ell=1}^{n} \sum_{k=1}^{m} \sigma_{a_\ell,b_k} \lambda_{a_\ell,i} \lambda_{b_k,j} \tag{11}$$

in Lemma 3.

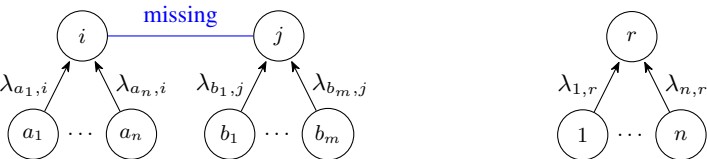

Figure 3: Left: A single missing edge on the top layer. Right: The gadget storing the value of each variable. $\lambda_{i,r}$ corresponds to $x_i$.

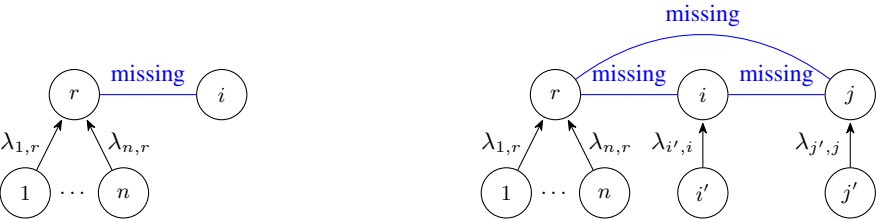

Figure 4: Left: Gadget for affine linear constraints. Right: Gadget for multiplicative constraints.

**Observation 4.** *All $\sigma$ values that appear in* (11) *cannot appear in any other missing edge equation of missing edges in the top layer. Parameters $\sigma_{a_\ell,j}$ can only appear in an equation of a missing edge that contains the node $j$ and another node $h$ such that there is a directed edge from $a_\ell$ to $h$. However, $(a_\ell, i)$ is the only such edge, since the nodes in the bottom layer only have outdegree one. The same is true for $\sigma_{b_k,j}$. $\sigma_{a_\ell,b_\ell}$ can only appear in an equation of a missing edge $h \leftrightarrow h'$ if there are directed edges $(a_\ell, h)$ and $(b_k, h')$. By the same argument, $i \leftrightarrow j$ is the only such missing edge. Furthermore $\sigma_{i,j}$ can obviously only appear in this equation.*

The above observation means that we can freely "program" the equations, that is, we can freely choose the $\sigma$-values in each missing edge equation without interfering with any other missing edge equation.

We start with a gadget with one node $r$ in the top layer and $n$ nodes in the bottom layer connected to it. It is used to store the value of each variable of our ETR instance, $\lambda_{1,r}$ corresponds to $x_1$, $\lambda_{2,r}$ corresponds to $x_2$, etc, see Figure 3 (right-hand side) for an illustration.

By assuming all polynomials in our $\text{QUAD}^{++}$-instance are of the forms (4), we need to be able to encode products and affine linear forms. We start by showing how to encode an arbitrary affine linear constraint $\sum_{\ell=1}^{n} \alpha_\ell x_\ell = \beta$ using a single additional node $i$ in the top layer, "connected" to $r$ via a missing edge as in Figure 4.

Setting $\sigma_{r,i} = \beta$ and $\sigma_{\ell,i} = \alpha_\ell$, $1 \leq \ell \leq n$, makes (11) together with $\lambda_{\ell,r} = x_\ell$ directly equivalent to $\sum_{\ell=1}^{n} \alpha_\ell x_\ell = \beta$.

Encoding a product $x_a = x_b \cdot x_c$ requires two additional nodes $i$ and $j$ in the top layer, with missing bidirectional edges between them and $r$. Furthermore we introduce two nodes $i'$ and $j'$ in the bottom layer, connected to $i$ and $j$ respectively, see Figure 4 (right-hand side). This introduces three equations. The missing edge $r \leftrightarrow i$ enforces $\lambda_{i',i} = \lambda_{c,r}$ by setting $\sigma_{r,i} = \sigma_{1,i'} = \ldots = \sigma_{n,i'} = 0$, $\sigma_{r,i'} = -1$, $\sigma_{c,i} = 1$, and $\sigma_{\ell,i} = 0$, for all $\ell \in \{1, \ldots, n\} \setminus \{c\}$ in (11). We use the missing edge $i \leftrightarrow j$ to further ensure $\lambda_{j',j} = \lambda_{i',i} = \lambda_{c,r}$, for which we set $\sigma_{i,j} = \sigma_{i',j'} = 0$, $\sigma_{i',j} = 1$, and $\sigma_{i,j'} = -1$. After having copied $\lambda_{c,r}$ twice, we are finally able to enforce the multiplication itself using the missing edge $r \leftrightarrow j$. We set $\sigma_{r,j} = \sigma_{r,j'} = 0$, $\sigma_{a,j} = 1$, $\sigma_{b,j'} = 1$, $\sigma_{\ell,j} = 0$, for all $\ell \in \{1, \ldots, n\} \setminus \{a\}$, and $\sigma_{\ell,j'} = 0$, for all $\ell \in \{1, \ldots, n\} \setminus \{b\}$. We need to copy the parameter $\lambda_{c,r}$ twice to be able to "program" the equation corresponding to the missing edge $r \leftrightarrow j$.

*Proof of Theorem 2.* Let polynomials $p_1, \ldots, p_m$ in variables $x_1, \ldots, x_n$ be a $\text{QUAD}^{++}$-instance with all polynomials being one of the forms in (4). Let the number of affine linear polynomials among $p_1, \ldots, p_m$ be $k$. Then the graph $G = (V, D, B)$ constructed above has $\ell := 1 + n + k + 4(m - k) = 1 + n + 4m - 3k$ nodes. Using Observation 4, we see that the construction induces a well-defined partial matrix $\Sigma \in \mathbb{R}^{\ell \times \ell}$. Every entry of $\Sigma$ not defined by the construction is set to

0 if it is off-diagonal and $\ell$ if it is on the diagonal. Since all $\sigma_{i,j}$ set in the construction are from $\{-1, 0, 1\}$ and off-diagonal, $\Sigma$ strictly diagonally dominant by our choice of $\ell$ and thus positive definite by the Gershgorin circle theorem [23].

Remains to prove $\Sigma \in \operatorname{im} \phi_G$. Let $\xi \in \mathbb{R}^n$ be any solution with $p_1(\xi) = \cdots = p_m(\xi) = 0$. The existence of $\xi$ is guaranteed by the promise of QUAD$^{++}$. Create $\Lambda \in \mathbb{R}^{\ell \times \ell}$ as follows: $\lambda_{i,r} = \xi_i$ for $i \in \{1, \ldots, n\}$ and $\lambda_{i',i} = \lambda_{j',j} = \xi_c$ whenever the vertices $i, i', j, j'$ are the vertices added by the construction due to a multiplication. All other entries of $\Lambda$ are 0. Then $I - \Lambda$ is invertible and we have $\Sigma = \phi_G(\Lambda, \Omega)$ for $\Omega = (I - \Lambda)^T \Sigma (I - \Lambda)$. Furthermore $\Omega$ is positive definite due to $\Sigma$ being positive definite and $\Omega \in \operatorname{PD}(B)$.

By Lemma 3, this implies that $|\mathcal{F}_G(\Sigma)|$ is precisely the number of solutions of our QUAD$^{++}$-instance. So if the QUAD$^{++}$-instance is a yes-instance, that is, has more than one solution, then our constructed instance is not numerically identifiable. If the QUAD$^{++}$-instance is a no-instance, that is, has only one solution, then our constructed instance is numerically identifiable. So we have a reduction from the complement of QUAD$^{++}$. The theorem now follows, since by Corollary 1, QUAD$^{++}$ is $\exists \mathbb{R}$-hard and the complement of an $\exists \mathbb{R}$-hard problem is $\forall \mathbb{R}$-hard. $\qquad \square$

## 5 Upper Bound for Numerical Identifiability

In this section, we show a $\forall \mathbb{R}$ upper bound for numerical identifiability and thus, combined with Theorem 2, prove an almost[3] matching lower and upper bound. We start with the following lemma. The $\exists \mathbb{R}$ part will be needed in the next section.

**Lemma 5.** *Membership in* $\operatorname{PD}(n)$ *and* $\operatorname{PD}(B)$ *can be expressed in* $\exists \mathbb{R}$ *and* $\forall \mathbb{R}$. [4]

*Proof.* For the $\exists \mathbb{R}$ expression, we use the fact that every real positive definite matrix $A \in \mathbb{R}^{n \times n}$ has a Cholesky decomposition $A = LL^T$ where $L$ is a real lower triangular matrix with positive diagonal entries. We can thus express $A \in \operatorname{PD}(n)$ as

$$\exists L \in \mathbb{R}^{n \times n} : A = LL^T \wedge \bigwedge_{i \in \{1, \ldots, n\}} (L_{i,i} > 0 \quad \wedge \bigwedge_{j \in \{i+1, \ldots, n\}} L_{i,j} = 0). \tag{12}$$

We quantify over matrices and consider matrix equations in (12). But this can be easily rewritten as an ETR-instance by quantifying over all entries of the matrix and having one individual equation for each entry of the matrix equation.

For the $\forall \mathbb{R}$ expression, we directly use the definition of positive definite matrices to express $A \in \operatorname{PD}(n)$ as

$$\forall x \in \mathbb{R}^n : x \neq 0 \implies x^T A x > 0. \tag{13}$$

For membership $A \in \operatorname{PD}(B)$, in both $\exists \mathbb{R}$ and $\forall \mathbb{R}$, we add the constraint $\bigwedge_{(i,j) \notin B \wedge i \neq j} A_{i,j} = 0$ to (12) and (13), respectively. $\qquad \square$

We remind the reader that numerical identifiability is a promise problem with the promise that the input $\Sigma \in \operatorname{im} \phi_G$, so it suffices to check whether all elements in the fiber $\mathcal{F}_G(\Sigma)$ are identical.

$$\forall \Lambda_1, \Lambda_2 \in \mathbb{R}^D, \Omega_1, \Omega_2 \in \operatorname{PD}(B) :$$
$$\phi_G(\Lambda_1, \Omega_1) = \phi_G(\Lambda_2, \Omega_2) = \Sigma \Rightarrow (\Lambda_1 = \Lambda_2 \wedge \Omega_1 = \Omega_2). \tag{14}$$

The checks $\phi_G(\Lambda_i, \Omega_i) = \Sigma$ are implemented using $\Omega_i = (I - \Lambda_i)^T \Sigma (I - \Lambda_i)$, which is equivalent due to $I - \Lambda_i$ being invertible for any $\Lambda_i \in \mathbb{R}^D$. This proves the following:

**Theorem 6.** *Numerical identifiability is in (the promise version of)* $\forall \mathbb{R}$.

**Remark 7.** *Strictly speaking, numerical identifiability is not contained in* $\forall \mathbb{R}$*, since it is a promise problem, that is, the outcome is not specified for $\Sigma$ that are not feasible. $\forall \mathbb{R}$ consists by definition only of classical decision problems, where the outcome is specified for* all *inputs. So the corresponding complexity class is* Promise-$\forall \mathbb{R}$. *Section B contains some more information on promise problems for the reader's convenience.*

---

[3]see Remark 7 for the details.

[4]Note that membership in $\operatorname{PD}(n)$ can be even decided faster. However, this will not change the complexity of our overall algorithm.

However, we can express feasibility in $\exists\mathbb{R}$:

**Lemma 8.** *Membership in* $\operatorname{im}\phi_G$ *can be expressed in* $\exists\mathbb{R}$.

*Proof.* We use the expression $\exists\Lambda\in\mathbb{R}^D,\Omega\in\operatorname{PD}(B):(I-\Lambda)^T\Sigma(I-\Lambda)=\Omega$, where we use Lemma 5 to express $\Omega\in\operatorname{PD}(B)$ in $\exists\mathbb{R}$, that is, we quantify over an arbitrary matrix $\Omega$ first and add the ETR expression from Lemma 5 to ensure that $\Omega$ is in $\operatorname{PD}(B)$. $\qquad\square$

Hence, we can check in $\exists\mathbb{R}$ whether the input $\Sigma$ is feasible and then in $\forall\mathbb{R}$ whether the fiber has only one element. Using Renegar's algorithm, we get:

**Corollary 2.** *Numerical identifiability can be decided in polynomial space.*

# 6 Generic Identifiability is in PSPACE

Let DIM denote the following problem: Given an encoding of a semi-algebraic set $S$ and a number $d$, decide whether $\dim S\geq d$.

**Lemma 9** (Koiran [27][5])**.** *The problem* DIM *is* $\exists\mathbb{R}$*-complete. Moreover, this is even true when the set is given by an existentially quantified formula as in (2).*

We use the same notation as in Section 2.1. Let $G=(V,D,B)$ be a mixed graph. Let $S_G=\{(\Lambda,\Omega)\mid|\mathcal{F}_G(\Lambda,\Omega)|>1,\Lambda\in\mathbb{R}^D,\Omega\in\operatorname{PD}(B)\}$.

**Observation 10.** $G$ *is generically identifiable iff* $\dim\mathbb{R}^D+\dim\operatorname{PD}(B)>\dim S_G$.

*Proof.* As $S_G\subseteq\mathbb{R}^D\times\operatorname{PD}(B)$ and $\dim(\mathbb{R}^D\times\operatorname{PD}(B))=\dim\mathbb{R}^D+\dim\operatorname{PD}(B)$, the right-hand side is just the definition of being generically identifiable. $\qquad\square$

We postpone the proof that membership in $S_G$ can be expressed in $\exists\mathbb{R}$, in favor of first giving our algorithm to decide generic identifiability, using this observation:

**Theorem 11.** *Generic identifiability is both in* $\forall\exists\mathbb{R}$ *and* $\exists\forall\mathbb{R}$.

*Proof.* Let $G$ be the given mixed graph. Formulate membership in $\mathbb{R}^D,\operatorname{PD}(B)$, and $S_G$ as instances of ETR using Lemmas 5 and 12. Note that the number of variables and the sizes of these instances are polynomial in the size of $G$. Now we can check whether $G$ is generically identifiable by checking the condition in Observation 10.

We first assume that we have oracle access to ETR, that is, we can query ETR a polynomial number of times. We decide whether $G$ is generically identifiable as follows:

1. Use Koiran's algorithm (see Lemma 9) repeatedly to compute $\dim S_G$ by checking whether $\dim S_G\geq d$ for $d=0,\ldots,2n^2$. (We could even use binary search.)

2. Compute $\dim\mathbb{R}^D$ and $\dim\operatorname{PD}(B)$ in the same way.[6] Here it suffices to check up to the maximum possible dimension of $d=n^2$.

3. Accept if $\dim\mathbb{R}^D+\dim\operatorname{PD}(B)>\dim S_G$, and reject otherwise.

The algorithm is correct by Observation 10. The algorithm above would already show the PSPACE upper bound for generic identifiability.

However, we can implement the algorithm above by a single formula by replacing the repeated use of Koiran's algorithm by a big disjunction:

$$\bigvee_{d_1=0}^{n^2}\bigvee_{d_2=0}^{n^2}\left(\dim\mathbb{R}^D\geq d_1\wedge\dim\operatorname{PD}(B)\geq d_2\wedge\dim S_G<d_1+d_2\right).$$

---

[5]see Appendix D for why this statement follows from [27].

[6]While we could compute these dimensions more directly, this is not necessary to obtain PSPACE algorithm. However for implementing this algorithm in practice, we would advise computing them more efficiently.

Note that the check $\dim S_G < d_1 + d_2$ needs to be implemented as $\neg(\dim S_G \geq d_1 + d_2)$, thus being in $\forall\mathbb{R}$ by De Morgan's laws. The existential and universal quantifiers are however independent, giving upper bounds of both $\forall\exists\mathbb{R}$ and $\exists\forall\mathbb{R}$. $\qquad\square$

Using Renegar's algorithm this implies:

**Corollary 3.** *Generic identifiability can be decided in* PSPACE.

It only remains to show how to express membership in $S_G$ as an ETR-formula.

**Lemma 12.** *Membership in $S_G$ can be expressed in* $\exists\mathbb{R}$.

*Proof.* The membership of some $(\Lambda, \Omega)$ in $S_G$ can be expressed as

$$
\Lambda \in \mathbb{R}^D \wedge \Omega \in \mathrm{PD}(B) \wedge \exists\Sigma \in \mathbb{R}^{n\times n}, \Lambda' \in \mathbb{R}^D, \Omega' \in \mathrm{PD}(B) : (I - \Lambda)^T\Sigma(I - \Lambda) = \Omega
$$
$$
\wedge (I - \Lambda')^T\Sigma(I - \Lambda') = \Omega'
$$
$$
\wedge (\Lambda \neq \Lambda' \vee \Omega \neq \Omega') . \qquad\square
$$

**Remark 13.** *The algorithm of Theorem 11 as is only tests generic identifiability. Since the problem has a high degree, one cannot expect that the solutions have easy expressions. However, Renegar's algorithm shows that the solutions are the linear factors of a certain polynomial, see [33].*

## 7  A Note on Cyclic Graphs

Our results so far have depended on the fact that every matrix $I - \Lambda$ with $\Lambda \in \mathbb{R}^D$ is invertible if the graph is acyclic. However, if the graph is cyclic, $I - \Lambda$ is not necessarily invertible. So in this case, we need to explicitly consider the subset $\mathbb{R}^D_{\mathrm{reg}}$ of matrices $\Lambda \in \mathbb{R}^D$ such that $I - \Lambda$ is invertible.

For matrices $\Lambda_0 \in \mathbb{R}^D_{\mathrm{reg}}$ and $\Omega_0 \in \mathrm{PD}(B)$, [21] define fibers as

$$
\mathcal{F}_G(\Lambda_0, \Omega_0) = \{(\Lambda, \Omega) \mid \phi_G(\Lambda, \Omega) = \phi_G(\Lambda_0, \Omega_0), \Lambda \in \mathbb{R}^D_{\mathrm{reg}}, \Omega \in \mathrm{PD}(B)\}.
$$

They determine generic identifiability for *cyclic* graphs in terms of these fibers. That is a mixed (cyclic) graph $G$ is said to be *generically* identifiable if $|\mathcal{F}_G(\Lambda_0, \Omega_0)| = 1$ for Zariski almost all $\Lambda_0 \in \mathbb{R}^D_{\mathrm{reg}}$ and $\Omega_0 \in \mathrm{PD}(B)$.

This is the same criterion used for acyclic graphs, except $\mathbb{R}^D$ has been replaced by $\mathbb{R}^D_{\mathrm{reg}}$ twice. Matrix invertibility can be easily expressed in $\exists\mathbb{R}$, using the definition of invertibility:

$$
A \text{ is invertible} \iff \exists B \in \mathbb{R}^{n\times n} : AB = I.
$$

Hence, all our upper bounds also hold for general graphs.

## 8  Conclusions

Due to double exponential runtime, the state-of-the-art algorithm for the generic identification problem is often too slow to solve instances of reasonable size. For example, García-Puente et al. [22] report that the runtime varies between seconds and 75 days for graphs with four nodes. An interesting topic for future work would be to implement the (theoretical) algorithm presented in our paper.

We have given a new upper on the complexity of generic identifiability, namely PSPACE. More precisely, we showed that it is in $\exists\forall\mathbb{R}$ and $\forall\exists\mathbb{R}$. This can be even improved to $\forall\mathbb{R}$. It is not necessary to express the dimension of $\mathbb{R}^D$ and $\mathrm{PD}(B)$ in terms of the theory of the reals, but they can be calculated directly, $\dim\mathbb{R}^D = |D|$ as well as $\dim\mathrm{PD}(B) = n + |B|$. In the light of our hardness proofs for the new notion of numerical identifiability, we conjecture that generic identifiability is hard for $\forall\mathbb{R}$, too.

## Acknowledgments and Disclosure of Funding

This research was supported by the Deutsche Forschungsgemeinschaft (DFG) grant 471183316 (ZA 1244/1-1).

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

## Appendix

## A Edge Identifiability

For edge identifiability, we obtain the same results as for identifiability itself.

### A.1 Hardness of Numerical Edge Identifiability

If we analyze the reduction in Theorem 2 in a bit more detail, we can use the same reduction to show

**Corollary 4.** *Numerical edge identifiability is $\forall\mathbb{R}$-hard.*

*Proof.* If instead of starting with an arbitrary $\mathrm{QUAD}^{++}$-instance, we start with a $\mathrm{QUAD}^{++}$-instance generated by Lemma 1 and Corollary 1. These instances have a distinguished variable $y$, such that there is always a single solution with $y = 1$ and possibly multiple solutions with $y = 0$. W.l.o.g. let $x_1$ be this distinguished variable. Following the reduction in Theorem 2, we construct a graph $G$ and a $\Sigma \in \operatorname{im} \phi_G$, such that the fiber $\mathcal{F}_G(\Sigma)$ is isomorphic to the solutions our $\mathrm{QUAD}^{++}$-instance. In particular the value of $\lambda_{1,r}$ in the elements of $\mathcal{F}_G(\Sigma)$ is exactly the value of $x_1$ in the solutions to the $\mathrm{QUAD}^{++}$-instance. Thus $|\mathcal{F}_G^{1,r}(\Sigma)| = 1$ iff the $\mathrm{QUAD}^{++}$ instance has exactly one solution, otherwise $|\mathcal{F}_G^{1,r}(\Sigma)| = 2$. $\qquad\square$

### A.2 Upper Bound for Numerical Edge Identifiability

Similarly to (14), we can express numerical edge identifiability as the $\forall\mathbb{R}$-formula

$$\forall \Lambda_1, \Lambda_2 \in \mathbb{R}^D, \Omega_1, \Omega_2 \in \mathrm{PD}(B) :$$
$$\phi_G(\Lambda_1, \Omega_1) = \phi_G(\Lambda_2, \Omega_2) = \Sigma \Rightarrow (\Lambda_1)_{i,j} = (\Lambda_2)_{i,j} . \tag{15}$$

This yields

**Theorem 14.** *Numerical edge identifiability is in (the promise version of) $\forall\mathbb{R}$.*

Again using Renegar's algorithm we also get

**Corollary 5.** *Numerical edge identifiability can be decided in polynomial space.*

### A.3 Generic Edge Identifiability is in PSPACE

We modify the algorithm of Section 6 to work with generic edge identifiability for some $\lambda_{i,j}$ rather than generic identifiability. Let $S_G^{i,j} = \{(\Lambda, \Omega) \mid |\mathcal{F}_G^{i,j}(\Lambda, \Omega)| > 1, \Lambda \in \mathbb{R}^D, \Omega \in \mathrm{PD}(B)\}$. Then we have the following analog to Observation 10:

**Observation 15.** *$\lambda_{i,j}$ is generically edge identifiable iff $\dim \mathbb{R}^D + \dim \mathrm{PD}(B) > \dim S_G^{i,j}$.*

**Lemma 16.** *Membership in $S_G^{ij}$ can be expressed in $\exists\mathbb{R}$.*

*Proof.* We use a similar formula to Lemma 12:

$$\Lambda \in \mathbb{R}^D \wedge \Omega \in \mathrm{PD}(B) \wedge \exists \Sigma \in \mathbb{R}^{m \times m}, \Lambda' \in \mathbb{R}^D, \Omega' \in \mathrm{PD}(B) : (I - \Lambda)^T \Sigma (I - \Lambda) = \Omega$$
$$\wedge (I - \Lambda')^T \Sigma (I - \Lambda') = \Omega'$$
$$\wedge (\Lambda_{i,j} \neq \Lambda'_{i,j})$$

$\qquad\square$

**Theorem 17.** *Generic edge identifiability is both in $\forall\exists\mathbb{R}$ and $\exists\forall\mathbb{R}$.*

*Proof.* We use the algorithm of Theorem 11, but replace the set $S_G$ by $S_G^{i,j}$. $\qquad\square$

**Corollary 6.** *Generic edge identifiability can be decided in* PSPACE.

# B   Promise problems

We give some background information on promise problems for the readers convenience. In a classical decision problem $L$, we are given an input and we have to decide whether $x \in L$ (the so-called yes-instances) or $x \notin L$ (the no-instances). For instance, in the classical SAT problem, we are given a Boolean formula $F$ in CNF. The yes-instances are the satisfiable formulas and the no-instances are the unsatisfiable one.

Promise problems have a third type of instances, the so-called do-not-care-instances. On these instances, an algorithm can do what it wants and give any output. For instance, consider the problem $\mathrm{SAT}^{++}$, where we ask the question of whether a satisfiable formula in CNF has another satisfying assignment. The yes-instances are all $F$ with at least two satisfying assignments, the no-instances are all $F$ with exactly one satisfying assignment, and the do-not-care-instances are all unsatisfiable $F$. An algorithm solving $\mathrm{SAT}^{++}$ has to output "yes" on every $F$ with at least two satisfying assignments and "no" on every $F$ with exactly one satisfying assignment. On unsatisfiable formulas, it can output whatever it wants. Note that every classical decision problem is also a promise problem with the do-not-care-instances being the empty set.

We can also define many-one reductions for promise problems: A function $f : \{0,1\}^* \to \{0,1\}^*$ is called a many-one reduction from a promise problem $L$ to another promise problem $L'$, if $f$ maps yes-instances of $L$ to yes-instances of $L'$ and no-instances of $L$ to no-instances of $L'$. $f$ can map do-not-care-instances of $L$ to any instance of $L'$. By using a similar trick of encoding an additional satisfying assignment like in the case of $\mathrm{ETR}^{++}$, one can show that $\mathrm{SAT}^{++}$ is NP-hard, since we can reduce SAT to it. This reduction maps the unsatisfiable formulas (no-instances of SAT) to formulas with one satisfying assignment (no-instances of $\mathrm{SAT}^{++}$) and satisfiable formulas (yes-instances of SAT) to formulas with two or more satisfying assignments (yes-instances of $\mathrm{SAT}^{++}$). Since SAT is a classical decision problem, there are no do-not-care-instances. $\mathrm{SAT}^{++}$ is, however, not contained in NP for formal reasons, because NP only contains classical decision problems.

# C   Semialgebraic sets

For the reader's convenience, we give a brief introduction to semialgebraic sets and discuss the notations important for this work. For details and proofs, we refer to the book [4].

A *semialgebraic set* in $\mathbb{R}^n$ is a finite Boolean combination (finite number of unions and intersections) of sets of the form $\{(x_1, \ldots, x_n) \mid f(x_1, \ldots, x_n) > 0\}$ and $\{(x_1, \ldots, x_n) \mid g(x_1, \ldots, x_n) \geq 0\}$. Here $f$ and $g$ are real polynomials in $n$ variables.

A *semialgebraic function* is a function $\mathbb{R}^n \to \mathbb{R}^{n'}$ with a semialgebraic graph, that is, the set of all $\{(x, f(x)) \mid x \in \mathbb{R}^n\}$ is a semialgebraic set.

From this definition of semialgebraic sets, it is easy to see that semialgebraic sets are the solutions of ETR-instances, that is, all $(x_1, \ldots, x_n)$ satisfying $\varphi(x_1, \ldots, x_n)$ in (2) form a semialgebraic set. From Tarski's theorem (see [4]), it follows that semialgebraic sets allow quantifier elimination, that is, all $(x_1, \ldots, x_n)$ satisfying

$$\exists y_1 \ldots \exists y_t \psi(x_1, \ldots, x_n, y_1, \ldots, y_t)$$

form a semialgebraic set, where $\psi$ (like $\varphi$) is a quantifier-free Boolean formula over the basis $\{\vee, \wedge, \neg\}$ and a signature consisting of the constants 0 and 1, the functional symbols $+$ and $\cdot$, and the relational symbols $<$, $\leq$, and $=$. $\psi$ depends on two sets of variables. It is clear that all $(x_1, \ldots, x_n, y_1, \ldots, y_t)$ satisfying $\psi$ form a semialgebraic set. Tarski's theorem tells us that we still get a semialgebraic set when we are projecting the $y_1, \ldots, y_t$ away using the existential quantifiers. This is used frequently in our proofs.

**Definition 4.** *A semialgebraic set $S$ has dimension $d$ if there exists a $d$-dimensional coordinate subspace such that the image of $S$ under the canonical projection onto this subspace has a nonempty interior and there is no such subspace of dimension $d + 1$.*

Above, by a $d$-dimensional coordinate subspace, we mean the subspace of $\mathbb{R}^n$ of all points $x$ such that $x_i = 0$ for $i \notin I$ for some subset $I \subseteq \{1, \ldots, n\}$ and $|I| = d$.

**Theorem 18** (see e.g. [6])**.** *Let $S$ be a semialgebraic set. Then its dimension as a semialgebraic set (as in Definition 4) equals the dimension of its Zariski closure as an algebraic set.*

## D  Koiran's algorithm

Koiran mainly works in the so-called BSS-model of real computation. In this model, one is also allowed to use arbitrary real constants in the algorithm as well as real inputs. In $\exists\mathbb{R}$, we only allow the constants $0$ and $1$ and the inputs are given by some binary encoding. However, Koiran also considers the bit model. [27, Theorem 6] proves the computational equivalence of DIM and 4FEAS in the bit model. Since 4FEAS is $\exists\mathbb{R}$-complete [35], this implies that DIM is $\exists\mathbb{R}$-complete. Note that at the time Koiran wrote his paper, the class $\exists\mathbb{R}$ was not formally defined and therefore, Koiran does not mention it explicitly.

For the moreover part, note that [27, Section 1.1] discusses the representations of semi-algebraic sets that are supported by his proof. There he mentions existentially quantified formulas explicitly.

