# OpenReview forum: "On the Complexity of Identification in Linear Structural Causal Models"
_NeurIPS.cc/2024/Conference — NeurIPS 2024 poster_

### Official Review · Reviewer_Gx9Z · 2024-07-08

**Soundness:** 3
**Presentation:** 2
**Contribution:** 2
**Rating:** 5
**Confidence:** 2

**Summary:**

The paper examines the computational complexity of Generic and Numerical Identifiability problems. It provides some reduction from these problems to R complete classes. Based on these results, the paper proposes an algorithm for Generic Identifiability, which improves the state-of-the-art double exponential time.

**Strengths:**

1- The paper proposed new findings on complexity of different various causal identifiability problems.

2- The results demonstrate that these problems can be decided in PSPACE.

3- The papers provided a comprehensive literature review in introduction.

**Weaknesses:**

1- The importance of results and their technical novelty are not clearly discussed.

2- The main part primarily comprises theorems proofs, which are not completely clear. It would be beneficial to move some parts to the appendix and include more intuition and discussion in the main part.

3- The paper lacks experimental results demonstrating the correctness of the proposed algorithm and the improvement in running time.

**Questions:**

1- Is it possible to conduct an experiment to compare your algorithm with previous algorithms?

2- Could you elaborate on the importance of the results and the novelty of proofs? The reductions seem very natural.

I recommend to revise Sections 4, 5, and 6 as the results are very compact and ambiguous. I could not follow very well.

**Limitations:**

Yes.

---

> ### Author Rebuttal · Authors · 2024-08-06
>
> Answer to questions:
>
> 1. At the moment, we consider the main contribution of the paper on the theoretical side. We are currently working on implementations.
>
>
> 2. We would like to emphasize that since the formulation of the parameter identification problem in linear SEMs in the 1960s, this problem has been intensively studied by econometricians and sociologists, and more recently by the ML, AI and statistics communities. The key importance of parameter identification (from observed data) is that the inferred parameters can be used to estimate causal effects, especially in cases where randomized controlled trials are not possible.
>
> However, despite decades of extensive research, no (complete) algorithm has been found that matches the general applicability of the Gröbner basis approach and thus runs faster than in doubly-exponential-time. Moreover, no non-trivial lower bound (hardness result) on the computational complexity of this problem has been established which would justify the high computational complexity of state-of-the-art algorithms. In our work, we give a novel algorithm that provides a constructive upper bound on the computational complexity of the generic identification problem, which significantly improves on the best-known doubly-exponential bound. Moreover, we give the first hardness results for numerical parameter identification in linear SEMs showing that it is hard for the complexity class ∀R.
>
> The proof technique provided in our paper is novel. In hindsight, the proofs in Sections 5 and 6 look easy, but no one made this connection to the existential theory of the reals as well as the result by Koiran before. To the best of our knowledge, this proof technique has not been used before for linear causal model analysis, which would be of interest to the NeurIPS community.
> Furthermore, the reduction in Section 4 is quite astonishing, since already SCMs with only two layers turn out to be hard.
>
> Regarding your recommendation to revise Sections 4, 5, and 6:
>
> We think that our proofs are quite clear. However, when writing for diverse communities, it is always challenging to balance between intuition and mathematical rigor. We thank you for your advice and will follow it by moving more technical parts to the appendix and adding more intuition.

---

> > ### Comment · Reviewer_Gx9Z · 2024-08-09
> >
> > Thank you for your response. I have raised my evaluation; however, I remain uncertain about the paper's contribution, as I am not too much familiar with this area.

---

### Official Review · Reviewer_gLJ2 · 2024-07-11

**Soundness:** 3
**Presentation:** 2
**Contribution:** 3
**Rating:** 5
**Confidence:** 2

**Summary:**

The submission studies the problems of generic parameter identification and numerical identification, both of which represent prominent tasks in causal analysis. The results include a PSPACE (i.e., single-exponential) algorithm for generic parameter identification, as well as ForallR-hardness and a single-exponential time algorithm for numerical identification. The submission also extends these results to the closely related setting of edge identifiability.

**Strengths:**

The studied problems are computationally challenging and seem to be relevant for the NeurIPS community.

**Weaknesses:**

-The restriction to recursive models comes abruptly and is not motivated or discussed in the submission at all. A number of questions are left completely unanswered, such as: Are recursive models common? Have they been studied before? Do they occur in practice? Do the results generalize beyond recursive models? Is there a specific reason one should restrict one's attention to these models?

-The definition of the problems studied do not seem written in sufficiently accessible language and assume that readers are already familiar with several specialized concepts. As one example, "Zariski almost all" and "Zariski closure" is never defined. The submission would also be much more accessible if it included at least some high-level examples of problem instances and solutions.

-In terms of writing, several of the results are established using only semi-formal language and without what I would consider sufficient rigor. For instance, the construction for the proof of Theorem 2 is described using inline examples and descriptions, but given the crucial importance of the construction for the proof I would have expected the core properties of the construction to be established and formalized via lemmas or claims.

-The results seem to be obtained primarily by cleverly combining known techniques and results, with little new insights required. In particular, the main results in Sections 5 and 6 essentially follow via direct encodings into appropriate fragments of the Theory of the Reals.

Minor comments:
-row 48: "using alone Sigma" should be "using Sigma alone"
-row 71: malformed sentence
-row 301: "So we can check in..." should be "Hence, we can check in..."
-row 340: "*the* generic identification problem". Also, missing full stop at the end of the footnote.

**Questions:**

No additional questions, but the authors are welcome to respond to the specific concerns raised above.

**Limitations:**

The limitations are adequately discussed.

---

> ### Author Rebuttal · Authors · 2024-08-06
>
> - Recursive (or, in graphical language, acyclic) models are fairly standard and most commonly used in causality. They are the basic structural causal models used in causal inference (e.g. in the famous Pearl's do-calculus, Markov-equivalence of causal models, etc.) and in causal structure learning (e.g., the well-known PC algorithm by Spirtes and Glymour or GES algorithm by Chickering learn from observed data acyclic models). Also (causal) Bayesian models are represented by such acyclic graphs. Furthermore, when restricting general causal models to linear ones (which are the subject of our work), recursive models are most commonly used. In particular the papers studying parameter identification in linear SEMs cited in our work [8, 10, 11, 14, 15, 24, 28, 29, 31, 36, 37, 38, 40, 41] assume recursive causal models. Since some recent works try to consider cyclic models as well, we have indicated in line 35 that our work assumes the standard model. Nevertheless, as we observed very recently, this restriction is not needed and our results can be generalized to non-recursive models. We will add an explanation in the final version.
>
> - Identification in linear structural causal models naturally reduces via Equation (10) in the paper to solving polynomial systems of equations over the reals. Therefore, tools from commutative algebra and real algebraic geometry are inherently needed. We agree that we should have given short inline descriptions of the terms you mention. We will do so and also add a more detailed appendix on the basic algebraic concepts used.
>
> - We hoped that it would be easier and more accessible if we develop our constructions in a textual manner. We find this quite hard to balance, since the audience of the paper is quite diverse. Other reviewers wanted more informal explanations and less proofs.  Note that the proof of Theorem 2 for instance is still a rigorous proof, it is a so-called "gadget-based" proof, which are used commonly. The two SCMs in Figure 4 are not just examples. They are our basic building blocks. They can be instantiated in such a way that they implement the equations given in (4), the basic building blocks of a QUAD^{++} instance. The gadget on the right-hand side implements the equation ab - c = 0, and the other gadget all other four equations with different instantiations of the $\sigma$-values. This is described right after Figure 4. Then one simply takes these basic building blocks and combines them as described in the proof of Theorem 2 to encode the whole QUAD^{++} instance. We will expand the explanation of the proof structure accordingly.
>
> - Large parts of our work are conceptual. The complexity of generic identification has been an open problem since the 1960s. In hindsight, the proof looks easy, but no one made this connection to the existential theory of the reals as well as the result by Koiran before. Moreover, no hardness results for identification were known before.
>
> - Thank you for pointing out the grammatical errors/typos, we will correct them.

---

> > ### Comment · Reviewer_gLJ2 · 2024-08-08
> >
> > Thank you for your clarifications and comments; they have helped alleviate some of my concerns.

---

### Official Review · Reviewer_Q8jB · 2024-07-11

**Soundness:** 4
**Presentation:** 2
**Contribution:** 3
**Rating:** 6
**Confidence:** 2

**Summary:**

This paper looks into the parameter identification problem in linear structural causal models using only observational data. Under the Gaussian linear SEMs, the paper provides a polynomial-space algorithm that runs in exponential time. The paper also provides the hardness of the problem.

**Strengths:**

In my knowledge, the methods and resutls are new. The paper investigated this traditional problem from computational complexity theory's view, which seems interesting to involved in this problem. The paper linked this parameter identification problems to the NP/coNP and PSPACE and then provide hardness and bounds based on it. Through I could not entirely follow the steps in the proof, but the paper seems theoritically sound.

**Weaknesses:**

- The algorithm has some assumptions about the model itself. For instance, the model assumes normal noises and (known) topological sorting. But this does not weaken the paper's contribution.
- The algorithm needs access to an oracle.
- The notations can be further explained to enhance the reading flow; maybe give a few sentence definitions when the notions first appear in the paper.

**Questions:**

- There should be a constraint on the diagonal of matrices in PD$(m)$.
- My major question is regarding the construct of $G$ in Section 4:
  - The left figure in Figure 3 has all $\lambda$ with subscript $r$. Isn't it $i$ for the left and $j$ for the right?
  - There seem to be some typos in equation (11) as $\lambda_{a_{\ell},i}$ appears in all three summations. Is this equation obtained by expanding equation (10)?
  - Can the authors explain Observation 4 a bit more? I am not quite sure why this is the case.
- The paper mentioned the algorithms in multiple parts. Does it mean the algorithm starts from line 320? And does this algorithm learn the final parameters or only give generic identifiability?
- I'd suggest having a link from Koiran’s algorithm to Appendix D.

**Limitations:**

The authors state all assumptions.

---

> ### Author Rebuttal · Authors · 2024-08-06
>
> Regarding weaknesses:
>
> - We agree that learning the graph structure of the causal model and/or the topological ordering of variables is a difficult and very challenging task. However, we would like to emphasize that learning causal structures and/or the topological orderings was not the subject of this work and, as it is commonly accepted in studies on the parameter identification in linear SEMs, we assume that such a structure is given as an instance of the problem. It is also common to assume that noises are normal.
>  We would like to point out that while these assumptions make the upper bounds easier in some sense, they make hardness results more difficult to prove. So they only work partly in our favor.
>
> - We would like to explain that our algorithm (in the proof of Theorem 11) does not need to have access to an oracle. To make the description of the algorithm modular and easy to understand, we first provide an algorithm having oracle access to ETR and next we explain that using Renegar’s algorithm the ETR queries can be computed in PSPACE (Corollary 3). Therefore, we only use them to make the statements more convenient. You can think of it as procedures that we call. The oracle calls can always replaced by actual algorithms. We will point this out more clearly.
>
> - We thank the reviewer for commenting on the need to add some supporting explanations to the designations.
>
>
> Regarding Questions:
>
> - Being positive definite has some implications on the diagonal, but depending on the definition, it is not always stated explicitly. For instance, in the first definition used in the proof of Lemma 5, there is some explicit restriction stated, while the second definition in the proof does not have such an explicit restriction. Since both definitions are equivalent, there is an implicit restriction in the second definition. Being diagonally dominant implies positive definiteness but the other direction is not true in general.
>
> - We apologize for the confusion and thank you for pointing this out. A few indices are not correct, see below for corrections. (We made some last minute changes and should not have done this.)
>
>   -- On the left-hand side in Figure 3, the r's should be i and j, respectively.
>
>   -- In the sum in the middle of (11), $\lambda_{a_\ell,i}$ should be $\lambda_{b_k,j}$. We are sorry for the typo. The equation is obtained from (10) by simply writing down the entries of the matrix product and looking at entry (i,j).
>
>   -- Obseration 4 follows from (10) and the fact the our graph has only two layers. Since i and j are both in the top layer, $\sigma_{i,j}$ can only appear in the equation in (10) that corresponds to entry (i,j) in the matrix. It can never appear multiplied with any $\lambda_{k,\ell}$, because then a least one of i or j needs to be in the bottom layer. The value $\sigma_{a_\ell,j}$ can only appear in an equation corresponding to a  missing edge $\{h,j\}$ that contains j and when there is a directed edge $(a_\ell,h)$. But there is only one leaving $a_\ell$ by construction and this edge ends in i, hence $h = i$. The same is true for $\sigma_{b_k,i}$. For $\sigma_{a_\ell,b_k}$ to appear in an equation given by a missing edge $\{g,h\}$, there have to be directed edges $(a_\ell,g)$ and $(b_k,h)$. Again by construction, $g = i$ and $h = j$.
>
> - The algorithm itself starts in line 320. But it calls subroutines that are described earlier. In step 1, we obviously call Koiran's algorithm. In step 2, we call the algorithm from Lemma 5 to decide membership in PD(B). In step 3, we use the algorithm from Lemma 12 as well as Koiran's algorithm to compute $\dim\ S_G$. We will state this more explicitly.
>
> - The algorithm decides whether the instance is generically identifiable. If it is, it will automatically also obtain the parameters, since the ETR solver computes them on the way.

---

> > ### Comment · Reviewer_Q8jB · 2024-08-12
> >
> > Thanks to the authors for the all the responses. They address most of my concerns. While I'm not entirely confident in this field, I would like to raise my score to a 6. Also, I suggest that in revised versions, the author can use the 1 extra page to further elaborate on the definition and terms.

---

### Official Review · Reviewer_WjE6 · 2024-07-12

**Soundness:** 3
**Presentation:** 2
**Contribution:** 3
**Rating:** 7
**Confidence:** 4

**Summary:**

This paper is a theoretical examination of the problem of parameter identifiability in structural equation models for mixed graphs (which has applications in causal modeling), significantly improving upon known upper bounds and establishing a hardness result for the problem's complexity.

**Strengths:**

**Originality**: The paper has originality (by using the existential theory of the reals to study complexity of parameter identifiability in probalistic graphical models), and this originality leads to an interesting new perspective and good results on the problem of study.

**Quality**:  The paper is of high technical quality (other than a few presumed typos).

**Clarity**: I find the proofs and technical writing to be rigorous and clear, and the positioning compared to previous complexity results is quite clear.

**Significance**: The paper should have at least moderate impact in the sub-areas Causal Inference and Graphical Models (and high impact in the sub-sub-area Algebraic Statistics). Specifically, this paper established foundational complexity results that have wide relevance in causal modeling. Additionally, the paper leads to natural follow-up work in the form of implementing the presented theoretical algorithm and approaching similar identifiability questions for other model classes from the same perspective.

**Weaknesses:**

The main weakness (other than those mentioned in the Questions and Limitations sections below) is the clarity of the non-technical writing. There is not much to guide the reader through the technical parts, so I worry non-experts will have too much difficulty to see the value of these theoretical results.

**Questions:**

My main questions are about the "causal" aspect of the work as well as implications for other model classes.

1. Don't all these results hold for Gaussian recursive mixed graph models generally instead of specifically causal ones? In other words, I don't see how any of the results make use of causal assumptions (like Reichenbach's common cause assumption or the causal Markov assumption) or interventional settings. To be clear, I don't think this is a bad thing—I just want to make sure I'm not missing some assumptions that limit the results.
2. Do these results apply immediately to Gaussian DAG models? Or only the upper bound? Or is there some fundamental difference in the DAG setup?

Now some technical questions:

3. In the equation on lines 33 and 129, shouldn't it be $\epsilon_j$ to match the $X_j$ on the left-hand side?
4. In lines 43,44 and again 154, 154, it's claimed that knowing the $\lambda$s is sufficient for recovering the $\omega$s, but one also needs to know/estimate $\Sigma$, right? While this is reasonably clear to an expert, it's not explicitly mentioned in those sentences.
5. In line 214, doesn't it rather reduce to $x_1 = \ldots = x_n = 0$?
6. Can the authors elaborate on the last sentence of the proof of Theorem 2, especially after "and" (lines 269, 270)? It could help to rephrase Corollary 1 in those terms or add an explicit Corollary 1'.
7. What is meant with "By standard simulation results,..." on lines 6, 7? I don't see any simulations results in the paper or supplement.

**Limitations:**

Limitations (such as restriction to the linear Gaussian setting and only having theoretical results) are clear from a detailed reading, but they could (and should) be more explicitly mentioned as limitations of the work in, e.g., the Intro.

---

> ### Author Rebuttal · Authors · 2024-08-06
>
> Answers to main questions:
>
> 1. The Gaussian recursive mixed graph models imply causal assumptions. The value (distribution) of each node is only affected by its parents and the bidirected edges. That yields Reichenbach's assumption, that if two nodes are not independent, there is either a direct cause (through their parents) or a common cause (through a bidirected edge). Some variant of the Markov assumption also holds if one considers the bidirected edges as parents.
>
> 2. There are different kinds of DAG models, with unobserved variables and without unobserved variables.
>
> The one without unobserved variables is trivially identifiable. It is equivalent to the Gaussian recursive mixed graph models without bidirected edges.
>
> The one with unobserved variables is more general than mixed graph models, so one can expect the lower bound to hold. The classic reduction of a mixed graph model to the DAG model is to replace any bidirected edge X <--> Y by a path through an "unobserved" node U, i.e. X <-- U --> Y. The covariance of U influences the covariances of X and Y. If the identification problem in the original mixed graph means solving equation system (1), then the reduced identification problem is solving an equation system (1') which is obtained from (1) by replacing each $\omega_{XY}$ with the product $\lambda_{XU}\lambda_{YU}$ where the solution must not contain any $\sigma$ involving the unobserved variables. Now a solution to (1) is also a solution to (1') after replacing the omegas (because (1) has no variables involving the unobserved variables).
>
>
> Answers to technical questions:
>
> 3. You are right about lines 33 and 129, the index is wrong. Thanks for pointing this out, we will correct it.
>
> 4. That is right, we assume (as do all the previous results that we cite) that $\Sigma$ is given. We will point this out more clearly in the mentioned lines.
>
> 5. That is true, thanks for pointing this out as well. We will correct this.
>
> 6. The classes $\forall \mathbb{R}$ and $\exists R$ are co-classes in the sense that when a problem $L \in \forall \mathbb{R}$, then its complement $\bar L$ is in $\exists R$ and vice versa. The complement of QUAD$^{++}$ is the following problem: Given a set of quadratic equations with at least one solution, is this solution the only solution. It is a $\forall \mathbb{R}$ complete problem. By Lemma 3, our constructed instance is identifiable iff the given satisfiable set of quadratic equations has exactly one solution. So we have a reduction from the complement of QUAD$^{++}$, which is $\forall \mathbb{R}$ hard, to Numerical Identifiability and therefore, Numerical Identifiability is also $\forall \mathbb{R}$ hard.
>
> 7. Standard simulation means that PSPACE is contained in EXPTIME. It is easy to see that an algorithm that is polynomially space bounded is also exponential time bounded (modulo some small technical details). It can be found in standard text books on complexity, like Papadimitriou, Computational Complexity, Pearson, 1993. We will add the reference.

---

> > ### Comment · Reviewer_WjE6 · 2024-08-08
> >
> > Thanks for the comprehensive response! I'm satisfied with the answers/explanations.
> >
> > I would suggest trying to incorporate your answers to 1. and 2. into the main text of the paper (abstract, intro, discussion, as appropriate) to make it easier for the broader causality and ML communities to understand the importance of your results (which would help to address the main weakness mentioned in my original review).

---

> > > ### Author Response · Authors · 2024-08-10
> > >
> > > Thanks for reading our paper so carefully and for the helpful feedback. We will follow your suggestion.

---

### Official Review · Reviewer_ToDP · 2024-07-14

**Soundness:** 3
**Presentation:** 3
**Contribution:** 3
**Rating:** 6
**Confidence:** 3

**Summary:**

The paper tries to attach an important theoretic problem: the complexity of identification in linear SEMs. The author proposed an algorithm for generic identification with exponential running time while previous the best algorithm is in double exponential running time.

**Strengths:**

1. The paper is well organized and presented. For a reader who is not very familiar with the complexity theory, the writer make the conception such as $\forall \mathbb{R}$ easy to understand.

2. The over all idea to connect the identification with ETR and QUAD is good and easy to follow.

**Weaknesses:**

Purely from complexity the current algorithm is better than state-of-the-art, however we do not have any knowledge about the constant computational overhead in the algorithm. When applying the theory in the paper to build a practical algorithm, this might also be important.

**Questions:**

See weakness.

**Limitations:**

Yes

---

> ### Author Rebuttal · Authors · 2024-08-06
>
> We agree it is not guaranteed that theoretically faster algorithms will be better in practice. We are working on an implementation of our algorithms. However, experience shows that drastic improvements on the theoretical running time typically come with similar improvements on the practical side. (But we admit that there are sometimes exceptions.)
>
> We would like to emphasize that since the formulation of the parameter identification problem in linear SEMs in the 1960s, this problem has been widely studied by econometricians and sociologists, and more recently by the ML, AI,  and statistics communities, but no (complete) algorithm has been found that matches the general applicability of the Gröbner basis approach and that runs faster than in doubly-exponential-time. For the first time, our work provides a new algorithm that states a constructive upper bound on the computational complexity of the generic identification problem which significantly improves the doubly-exponential bound of Gröbner bases.
>
> We decided to publish our results in its current theoretical form because (1) it shows (for the first time) that it is possible to avoid the method based on Gröbner bases and (2) it proposes a new approach (using semi-algebraic sets and Koiran's algorithm) to solve the identification problem, which would be of interest to the NeurIPS community. We believe that our results also provide an important first step towards achieving an implementable algorithm that can handle cases with sizes much larger than those possible using state-of-the-art methods (see, e.g., Garcia-Puente et al. [22]).
>
> For the related problem of numerical identifiability, we get an almost tight complexity classification. A $\forall \mathbb{R}$ lower bound, which is to our knowledge the first lower bound for any such identification problem, and a(n almost) matching upper bound, namely promise $\forall \mathbb{R}$ or $\forall \exists \mathbb{R}$.

---

> > ### Comment · Reviewer_ToDP · 2024-08-12
> >
> > Thanks for the rebuttal, I believe the rebuttal resolved my concern.

---

### Author Rebuttal · Authors · 2024-08-06

- We thank the reviewers for their helpful comments. We make some remarks here addressing questions that were raised by more than one reviewer. Specific questions of the referees are discussed in the individual answers.

- We would like to stress that the identification in structural causal models is an old problem (at least for ML/AI standards) with a long history. Despite all the efforts, only very little is known about the computational complexity of identification. We are not aware that any general hardness results. Furthermore, the currently best running time of a complete algorithm for generic identification is doubly exponential. It is based on Gröbner bases and is almost 15 years old. We here obtain the first theoretical improvement since then by giving an algorithm in PSPACE, which implies an exponential improvement in running time. Moreover, we complement this with an almost matching lower bound for a related identification problem (Numerical identification).

---

### Comment · Area_Chair_4eg6 · 2024-08-10

Dear Reviewers,

The authors have submitted the point-to-point response.
Thankfully, many reviewers are engaged in the discussion.
Could ToDP and Q8jB please acknowledge that you have read the rebuttal?
Are there still some unresolved issues?

AC

---

### Decision · Program_Chairs · 2024-09-25

**Decision:**

Accept (poster)

**Comment:**

This paper addresses the complexity of parameter identification in linear Structural Equation Models (SEMs), introducing a new algorithm that significantly improves upon the previously known double-exponential time algorithms. The paper’s contributions include both theoretical advancements, such as establishing an upper bound on computational complexity, and the first hardness results for numerical identification in linear SEMs. While reviewers acknowledged the importance of these results, they also noted areas for improvement, including the need for clearer explanations and more intuitive presentations. Given the strong technical contributions are strong, the paper is recommended for acceptance. The authors are suggested to improve the clarity of writing.